# EasyQuant: An Efficient Data-free Quantization Algorithm for LLMs

**Hanlin Tang**
ranchotang@tencent.com

**Yifu Sun**
yifusun@tencent.com

**Decheng Wu**
woodchenwu@tencent.com

**Kai Liu**
raccoonliu@tencent.com

**Jianchen Zhu**
dickzhu@tencent.com

**Zhanhui Kang**
kegokang@tencent.com

## Abstract

Large language models (LLMs) have proven to be very superior to conventional methods in various tasks. However, their expensive computations and high memory requirements are prohibitive for deployment. Model quantization is an effective method for reducing this overhead. The problem is that in most previous works, the quantized model was calibrated using a few samples from the training data, which might affect the generalization of the quantized LLMs to unknown cases and tasks. Hence in this work, we explore an important question: *Can we design a data-free quantization method for LLMs to guarantee its generalization performance?*

In this work, we propose EasyQuant, a training-free and data-free weight-only quantization algorithm for LLMs. Our observation indicates that two factors: outliers in the weight and quantization ranges, are essential for reducing the quantization error. Therefore, in EasyQuant, we leave the outliers (less than 1%) unchanged and optimize the quantization range to reduce the reconstruction error. With these methods, we surprisingly find that EasyQuant achieves comparable performance to the original model. Since EasyQuant does not depend on any training data, the generalization performance of quantized LLMs are safely guaranteed. Moreover, EasyQuant can be implemented in parallel so that the quantized model could be attained in a few minutes even for LLMs over 100B. To our best knowledge, we are the first work that achieves comparable performance with data-dependent algorithms under a data-free setting and our algorithm runs over 10 times faster than the data-dependent methods.

## 1 Introduction

Recent work has already proved the superior performance of Transformer (Vaswani et al., 2017) based LLMs (Workshop, 2023; Zhang et al., 2022; Touvron et al., 2023; Brown et al., 2020; Rae et al., 2021; Smith et al., 2022; Chowdhery et al., 2022; Zeng et al., 2022) on various tasks over traditional methods, and has attracted massive interest in how to improve and utilize those LLMs. However, the model size also grows dramatically along with improved performance. Hence the memory footprint and computational cost become the bottleneck for deploying those models. One promising solution to alleviate this overhead is model quantization (Frantar et al., 2023a; Xiao et al., 2023), where we quantize weight only or weight and activation both i order to reduce memory consumption and computational cost.

Although model quantization is a well-studied area for normal-sized models, such as BERT (Devlin et al., 2018) and GPT-2 (Radford et al., 2019), it is still a quite challenging task for LLMs. One major reason is that previous lossless model quantization algorithms require retraining for the quantized model, which is too expensive for models over billions of parameters. Beyond this, previous models are usually designed for specific domain tasks, which means the training data are sampled from limited task domains. However, recent LLMs are usually trained on various domains of data corpus, and they have shown to be quite effective for multi-domain zero-shot tasks. In this case, if we only retrain the quantized LLMs using partial domain corpus, the generalization ability of LLMs might get worse. Therefore both efficiency and generalization guarantees are very important for designing LLMs quantization algorithms. To date, for low-bits weight-only quantization, several post-training algorithms have been proposed (Frantar et al., 2023a; Yao et al., 2022). However, those methods also require a small calibration set sampled from training data, which still takes at least several hours. Moreover, the use of those calibration data also brings the risk of making the model overfit to the calibration set.

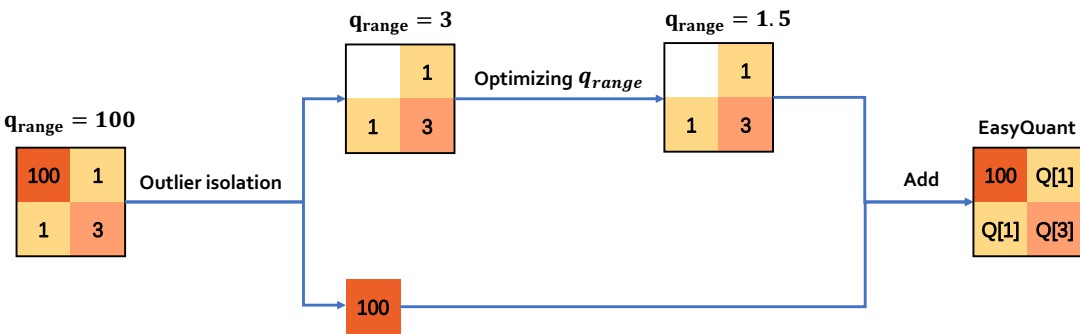

Figure 1: Pipeline of EasyQuant. We first find all the outliers in weight and keep them in full precision (fp32/fp16/bf16). Afterward, we optimize the quantization range (denoted as $q_{range}$) in order to approximate the normal values more precisely. In the end, the normal values are quantized into lower bits (denoted as $Q[\cdot]$) with optimized quantization ranges and we set the outliers unchanged in weight.

**Our Contribution:** In this work, we propose a novel data-free model quantization algorithm, namely EasyQuant, that potentially improves the performance of low-bits quantized LLMs. The generalization ability of LLMs is inherently guaranteed since EasyQuant does not need any input data. By running EasyQuant for only a few minutes, we can quantize public-available OPT-176B, BLOOM-176B, and LLAMA-65B into lower bits without significant loss on various benchmarks. To our best knowledge, this is the first data-free LLM quantization algorithm for LLM quantization without notable system overhead.

Moreover, our work reveals the essential factors that cause the performance degradation of the quantized LLMs. We show that the outliers in weights are more critical to the model's performance compared to the normal elements. Beyond this, we propose to use a gradient-based method for optimizing the quantization range. These two strategies can also be used in other scenarios, such as weight-activation quantization and quantization-aware training (QAT).

Last but not least, we develop efficient CUDA kernels for outlier isolation in dequantization, and proved that hold $1\%$ outliers in weights unquantized brings negligible (less than $0.1\%$) overhead w.r.t to overall latency. We also propose to implement EasyQuant in parallel for quantizing each weight in the model, which means a 175B-sized model can be quantized into 4-bits within 10 minutes.

## 2 Background and Motivation

The most widely used quantization method, namely rounding to nearest-number (**RTN**), quantizes a tensor $x$ into $k$-bits representation according to

$$Q[\boldsymbol{x}] = s \times \left\lfloor \text{clamp}\left(\frac{\boldsymbol{x}}{s}, l_{\min}, l_{\max}\right) \right\rceil . \quad (1)$$

Here $s$ is the quantization scale, $l_{\min}$ and $l_{\max}$ are the lower and upper bound for clipping, and $\lfloor \cdot \rceil$ is the rounding operator. Usually we set $l_{\min} = \left(-2^{k-1} + 1\right)$ and $l_{\max} = 2^{k-1}$ and set $s$ to be the maximum absolute value in $x$.

There are two major directions for finding the best configuration in weight-only LLM quantization. The first is to minimize the reconstruction error of the weight parameter (denoted as $W$), which is defined as

$$r(W) := \|Q[W] - W\|^2.$$

Notice that in this case we only need to have access to the weight itself, therefore it is data-free.

Beyond this, recent studies ([Frantar et al., 2023a](); [Yao et al., 2022]()) propose to use the output error, defined as

$$e(W) = \sum_{X \in \mathcal{D}} \|Q[W]X - WX\|^2 ,$$

where $\mathcal{D}$ is a calibration set sampled from the original training data, for optimization. This regulation tries to mimic the outputs from the original model directly hence achieving a more promising result than reconstruction-based methods.

**Data-dependent calibration might weaken the generalization ability of LLMs** However, the performance gain from using calibration data might jeopardize the generalization of the quantized model, because it brings the risk of making the model overfit to the calibration set. For example,

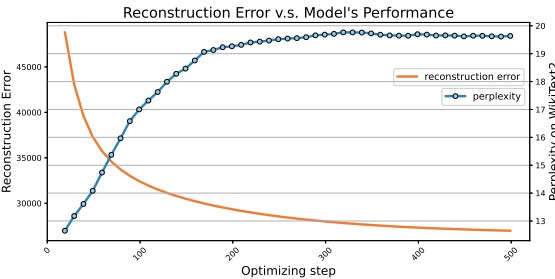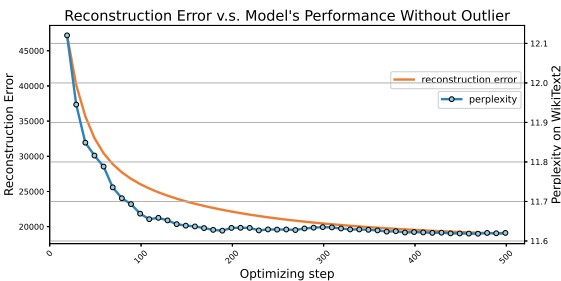

Figure 2: Smaller reconstruction error cannot guarantee a better model performance. Straightforwardly shrinking the quantization ranges will clip most of the outliers to be very small, hence the perplexity increases severely since those outliers are critical for preserving the model's performance. However, when keeping those outliers unquantized, the quantized model achieves a better performance as the reconstruction error decreases continuously. This result clearly suggests that the outliers are more important than the normal values in weight, and optimizing the quantization ranges using gradient defined in (2) can significantly increase the accuracy of quantized models. More details about the experiment can be found in Section 5.

both ZeroQuant and GPTQ involve changing the original weight by training or OBS in order to minimize the output error, therefore the distribution of the weight's parameters might deviate from the original. Since the calibration data is usually sampled from a few specific domains, the performance of the calibrated model on other tasks may not be guaranteed.

**Data-free quantization is challenging, but very important**    Although it's more challenging to use the reconstruction error as a regulation because it can only optimize the quantized model indirectly, still it is a very important direction for researching because the generalization ability of the model is inherently guaranteed when using data-free quantization since it uses no training data. Therefore in this paper, we aim to answer the following question:

*How can we efficiently recover the performance of the quantized model without using any input data?*

In this work we propose EasyQuant, a data-free fast algorithm that could significantly improve the performance of quantized LLMs in a data-free setting, and more importantly, even outperforms the results from data-dependent quantization algorithms. Our experiments reveal that the performance gap of the lower bits (e.g. 4-bits) quantized LLMs origins from two factors:

1. Setting the quantization range as the maximum absolute value of the weight induces a large reconstruction error for low-bits quantization.

2. The outliers in the weight matrix, which ac-

count for less than $0.1\%$ of the parameters, impose a very important influence on the model's performance.

In EasyQuant, we use quantization range minimization and outlier isolation to address these two challenges, and our results prove that EasyQuant achieves a significant improvement over RTN.

## 3    Insight behind EasyQuant

As mentioned above, the weight's outliers and quantization ranges are essential to the quantized model's performance. Below we present the supporting experiments in detail.

### 3.1    The quantization range can be efficiently optimized using gradient

Although the quantization operation itself is non-differentiable, the gradient of the reconstruction error ($\|Q[\boldsymbol{x}] - \boldsymbol{x}\|^2$) w.r.t. the quantization range $s$ is differentiable in most cases. We proved that the gradient of the quantization range $s$ admits (see Section 4 for more details)

$$\frac{\partial \|Q[\boldsymbol{x}] - \boldsymbol{x}\|^2}{\partial s} = 2 \sum_i \left( (Q[x_i] - x_i) \left\lfloor \frac{x_i}{s} \right\rceil \right).$$
(2)

With this gradient, the reconstruction error can be quickly minimized within hundreds of steps (see Figure 2 for more details). This result indicates that by shrinking the quantization range, most of the parameters in weight can be approximated more precisely. However, as shown in Figure 2, the performance of the quantized weight gets even worse

as the reconstruction error decreases. This is a very counter-intuitive result.

Through in-depth analysis, we realized that when decreasing the quantization range, more salient parameters outside the quantization range would be clipped out. Although most of the weights get approximated more precisely as indicated by the decreased reconstruction error, the salient parameters are poorly represented. As the model performance drops severely in this case, we realized that those outliers are way more important than the normal elements for the model's performance.

### 3.2 Outliers in weight are very important, but not sufficient

Before we further discuss the influence of those outliers, we first provide a $(n\sigma)$ criterion for defining the outliers in weight. For any weight $W$, we say its $(i, j)$-th number $W_{i,j}$ is an $(n\sigma)$ outlier if

$$|W_{i,j} - mean(W)| \geq n * var(W), \quad (3)$$

where $mean(W)$ and $var(W)$ are the mean and variance of $W$.

Now the question is: *Can we hold those outliers unchanged and straightforwardly compress the normal elements into lower bits?* Unfortunately, our result suggests that excluding the outliers from quantization solely is not enough. As shown in Table 1, the performance gap still exists even when we hold $1\%$ numbers in fp16. The problem is that if we keep too many numbers in fp16, the overhead of the dequantization kernel would also increase and result in a decreased overall throughput.

### 3.3 EasyQuant potentially improve the performance

As shown in Section 3.1 and Section 3.2, optimizing the quantization ranges directly reduces the model's performance drops severely because of the clipped outliers. These key observations inspire us to design EasyQuant, in which we isolate the outliers from quantization first and then optimizing the quantization range for the remaining elements. As shown in the right part of Figure 2, with outliers being kept unquantized, the performance of the quantized model increases continuously under decreased reconstruction. This clearly proves we can potentially improve the performance of quantized LLMs with this strategy.

## 4 Methodology

### 4.1 Driving of the gradient in (2)

Let's say the original scale $s$ gets an infinitely small variation $\Delta s$, which means

$$\left\lfloor \frac{x}{s + \Delta s} \right\rfloor = \left\lfloor \frac{x}{s} \right\rfloor, \quad \text{if } \frac{x}{s} - \left\lfloor \frac{x}{s + \Delta s} \right\rfloor \neq 0.5.$$

Therefore we get

$$
\begin{aligned}
Q_{s+\Delta s}[x] &= (s + \Delta s) \left\lfloor \frac{x}{s + \Delta s} \right\rfloor \\
&= (s + \Delta s) \left\lfloor \frac{x}{s} \right\rfloor,
\end{aligned}
$$

this leads to

$$\frac{\partial Q[x]}{\partial s} = \frac{Q_{s+\Delta s}[x] - Q_s[x]}{\Delta s} = \left\lfloor \frac{x}{s} \right\rfloor.$$

This gives us

$$
\begin{aligned}
&\frac{\partial \|Q[\boldsymbol{x}] - \boldsymbol{x}\|^2}{\partial s} \\
=& 2 \left\langle Q[\boldsymbol{x}] - \boldsymbol{x}, \frac{\partial Q[\boldsymbol{x}]}{\partial s} \right\rangle \\
=& 2 \left\langle Q[\boldsymbol{x}] - \boldsymbol{x}, \left\lfloor \frac{x_i}{s} \right\rfloor \right\rangle \\
=& 2 \sum_i \left( (Q[x_i] - x_i) \left\lfloor \frac{x_i}{s} \right\rfloor \right).
\end{aligned}
$$

### 4.2 Algorithm description

In EasyQuant, for each weight $W$, we first select all $(n\sigma)$ outliers (using (3)) and store its index $I^o(W)$. Afterward, for the normal elements, we optimize the per-channel quantization range using an optimizer (in our case we use Adam for example) with gradients defined in (2). The final quantized weight from EasyQuant can be formulated as

$$
\begin{aligned}
&Q^{EasyQuant}[W] \\
=& Mask^o(W) * W + (1 - Mask^o(W)) * Q[W],
\end{aligned}
\quad (4)
$$

where $Mask^o$ is a mask tensor defined as

$$Mask^o_{i,j}(W) = \begin{cases} 1 & \text{if } (i, j) \in I^o(W), \\ 0 & \text{if } (i, j) \notin I^o(W). \end{cases} \quad (5)$$

The detailed description of EasyQuant is in Algorithm 1.

| Threshold $n$ (BLOOM-7B) | Baseline | 1 | 2 | 4 | 6 |
|---|---|---|---|---|---|
| PPL on WikiText2 | 11.37 | 12.153 | 12.495 | 12.518 | 12.536 |

Table 1: Isolating outliers in weight from quantization can increase the model's performance. Here $n$ refers to the hyper-parameter in the outlier criterion ($n\sigma$) as defined in (3) and baseline is the result from unquantized model. Notice that even with $10\%(n = 1)$ numbers being held unquantized, there is still a large gap to the baseline. This means isolating the outliers is not enough to fully recover the accuracy of quantized models.

---

**Algorithm 1** EasyQuant

1: **Initialize**: outlier threshold $n$, hyper-parameters for optimizer $\mathcal{A}$, original weight $W$.
2: **Quantize:**
3:     According to (3), compute the index $I^o(W)$ of the ($n\sigma$) outliers in $W$.
4:     Optimizing the quantization range $s$ using optimizer $\mathcal{A}$ with gradient defined in (2).
5:     Quantize $W$ into $Q[W]$.
6: **Dequantize:**
    $Q^{EasyQuant}[W] = Mask^o(W) * W + (1 - Mask^o(W) * Q[W]$, where $Mask^o(W)$ is defined in (5).

---

## 5 Experiment

**Baselines:** We compare EasyQuant with several baselines in the INT4 quantization setting below:

- **RTN**: The model's weights are naively quantized according to (1).

- **ZeroQuant**: The algorithm proposed in Yao et al. (2022). Authors treat each layer as a small neural network and use the original as the teacher model to distill the quantized one. This is equivalently minimizing $\sum_{\boldsymbol{x} \in \mathcal{D}} \|f(W^T; \boldsymbol{x}) - f(W^S; \boldsymbol{x})\|^2$ where $x$ are the input activations, $W^T$ is the weight of the original model and $W^S$ is the quantized model.

- **GPTQ**: This algorithm is proposed in Frantar et al. (2023a). Authors use the same objective function $\sum_{\boldsymbol{x} \in \mathcal{D}} \|f(W^T; \boldsymbol{x}) - f(W^S; \boldsymbol{x})\|^2$ as in ZeroQuant. But they utilize OBS for minimizing the loss function instead of using a gradient-based optimizer.

**Experiment Setup.** For all models, we set the outlier threshold $n \in [2.5, 3]$ in order to ensure that the outliers account less than $1\%$ of all numbers. For BLOOM and LLAMA, we use $n = 3$. When optimizing the quantization ranges, we use Adam as the optimizer and set the learning rate $1e - 3$ for BLOOM and $1e - 4$ for LLAMA. We choose the quantization ranges from step 100 for BLOOM

and 500 for LLAMA. We use symmetric quantization since the normal values are symmetrically distributed with the outliers being excluded. For a fair comparison, we use per-channel quantization for weight in all algorithms (which means each column shares one common quantization range).

**Evaluation Tasks.** As for the evaluation tasks, we mainly focus on perplexity-based tasks, as they are known to be particularly sensitive to model quantization Frantar et al. (2023b). The perplexity tasks we include are WikiText2 (Merity et al., 2016), Penn Treebank (Marcus et al., 1994) and C4 (Raffel et al., 2020). The zero-shot tasks' results are also provided, such as PIQA (Tata and Patel, 2003), ARC (Boratko et al., 2018) and StoryCloze (Mostafazadeh et al., 2017).

**Implementation.** Since each weight can be quantized in parallel, therefore we use $8* $ A100 for running EasyQuant, and we finish the quantization in $1 \sim 10$ mins for all models. We store the index and value for all outliers together with the quantized normal values. Our dequantization kernel is built using CUDA.

### 5.1 Experiment Analysis

We focus our study on LLM by quantizing the entire BLOOM, and LLAMA model families to 4-bit.

**Perplexity-base tasks.** We first study perplexity-based tasks. On LLaMA models, Table 2 shows that EasyQuant outperforms GPTQ in most cases. For LLaMA-65B, GPTQ drops 4.21 points on PTB, performing worse than the $9 \times$ smaller full-precision 7B model, while EasyQuant still performs well on this task. On the other tasks, EasyQuant losing only 0.4–0.7 points. BLOOM shows a similar pattern (see Table 10 in appendix): EasyQuant drops only 0.1-0.16 points on perplexity-based tasks. Notice that we observe a smaller gap between our method and GPTQ on C4. It is mostly because, as a data-calibrated quantization method, GPTQ uses C4 dataset for calibra-

| | | Perplexity-based Task | | | | | Perplexity-based Task | | |
|---|---|---|---|---|---|---|---|---|---|
| | | WikiText2 | PTB | C4 | | | WikiText2 | PTB | C4 |
| LLAMA–7B | fp16 | 5.68 | 8.80 | 7.08 | LLAMA–33B | fp16 | 4.10 | 7.30 | 5.98 |
| | RTN | 6.29 | 11.25 | 8.12 | | RTN | 4.54 | 8.65 | 6.54 |
| | GPTQ | 6.09 | 11.56 | 7.78 | | GPTQ | 4.45 | **8.44** | 6.40 |
| | EasyQuant | **6.01** | **10.72** | **7.71** | | EasyQuant | **4.34** | 8.45 | **6.37** |
| LLAMA–13B | fp16 | 5.09 | 8.07 | 6.61 | LLAMA–65B | fp16 | 3.53 | 6.91 | 5.62 |
| | RTN | 5.53 | 9.77 | 7.23 | | RTN | 3.99 | 10.67 | 6.45 |
| | GPTQ | 5.36 | 9.49 | 7.07 | | GPTQ | 4.13 | 11.12 | 6.38 |
| | EasyQuant | **5.29** | **9.37** | **6.97** | | EasyQuant | **3.98** | **9.61** | **6.30** |

Table 2: Perplexity results for LLAMA model family

tions.

**Zeroshot tasks.** For most zero-shot tasks, EasyQuant achieves harmless performance with only 0.1 %-0.52% accuracy drops as shown in Table 10 in appendix and outperforms GPTQ on most cases. Here we simply use the implementation of GPTQ on LLAMA from its git.[1] We note that EasyQuant can be further improved via finer-granularity grouping. However, we will not include this overhead in this paper.

| outlier ratio | overhead |
|---|---|
| 0.01% | 0.027ms |
| 0.10% | 0.055ms |
| 0.50% | 0.093ms |
| 1% | 0.117ms |
| 5% | 0.186ms |
| 10% | 0.212ms |

Table 3: Overhead of outlier isolation on A100

**Practical Latency.** We evaluate the overhead of EasyQuant by comparing the overhead of outlier isolation, int4 dequantization, and matrix multiplication with batch size 1, sequence length 1024, on a single A100 GPU. The matrix size is $14336 \times 53746$ which is the same as the first FFN layer in 176B BLOOM. For outlier isolation, we test the latency of outliers ratio (fraction of outliers within the weight) in 6 settings: $(0.01\%, 0.10\%, 0.50\%, 1\%, 5\%, 10\%)$. The matrix multiplication takes 83ms and dequantization takes 5ms. Therefore from Table 3 we can see that recovering the outliers in weight brings almost no overhead to the overall latency.

**Ablation study.** To understand the effect of unstructured outliers, we show the perplexity result of EasyQuant without outlier isolation or quantization

[1]https://github.com/qwopqwop200/GPTQ-for-LLaMa

range optimization. As discussed in Section 3, both strategies impose a very important influence on the final model performance.

We further conduct experiments proving whether the performance gain mainly comes from the outlier isolation: Actually, outlier isolation is a very important component of EasyQuant, but still not enough to fully recover the performance loss from quantization. Keeping even 10% of weights as fp16 outliers still admits about 8% ppl increase while EasyQuant admits only 1% ppl increase. Below we present the result of 4-bit quantized BLLOM-7B when we just keep 1% outliers in fp16 without quantization range optimization on various benchmarks.

| Benchmark | EasyQuant | 1% fp16 outlier |
|---|---|---|
| WikiText2(PPL) | 11.66 | 12.52 |
| PTB (PPL) | 21.42 | 23.32 |
| C4(PPL) | 15.46 | 16.44 |
| PIQA (ACC) | 73.61% | 72.74% |

Table 4: Using outlier isolation solely is not enough to fully recover the performance loss. EasyQuant consistently outperforms outlier isolation in all benchmarks.

**Outlier influence.** The outlier isolation is a key component in EasyQuant, but it can only impose an indirect influence on the model accuracy. The interesting phenomenon we find is that the outliers behave like a gating mechanism: without outlier isolation, the model achieves a much worse performance under a small reconstruction error; however, when keeping those outliers in fp16, the quantized LLM attains a continuously decreased ppl under smaller reconstruction error:

Moreover, we have also conducted a complementary experiment testing the direct influence of the weight outlier: We prune 1% of the values ( according to its magnitude) in weights into 0 and see the ppl results (as shown in Table 6). It has

| reconstruction error | int4 outlier | fp16 outlier |
|---|---|---|
| 4.8E4 | 12.65 | 12.50 |
| 3.5E4 | 14.73 | 11.61 |
| 2.7E4 | 19.71 | 11.25 |
| 2.3E4 | NA | 11.10 |
| 1.9E4 | NA | 11.02 |

Table 5: ppl results on Wikitext2 of BLOOM-7B with and without outlier isolation.

shown that the largest value (outliers) imposes the same influence on the model performance as the normal values (median), which means those outliers share the same direct influence on the model accuracy with normal values. Therefore outlier isolation imposes a key influence on the model accuracy indirectly.

| pruned weights | PPL |
|---|---|
| smallest (top-0% 1%) | 11.66 |
| median (top-49% 50%) | 19.16 |
| largest (top-99% 100%) | 19.17 |

Table 6: ppl results after pruning 1% weight with different magnitude

**Outlier distribution.** We also explore the outlier distribution along different modules and layers. It shows that the fraction of outliers shares different patterns in different modules and layers (as shown in Table 7 and 8). FFN.2 has a significantly higher fraction of outliers. However, it shows no pattern along the layer index.

| module name | outlier fraction (%) |
|---|---|
| Att.qkv | 0.2993 |
| Att.output | 0.5036 |
| FFN.1 | 0.288 |
| FFN.2 | 0.7560 |

Table 7: Outlier fraction distribution in different modules in BLOOM-7B under 3-sigma threshold

**Quantization range.** The dynamic of the quantization range is shown in Table 9. Roughly speaking, this range decreases fast in the early stage of training, which means a smaller quantization range will make most of the parameters to be quantized more precisely. After certain steps of training, the quantization range becomes stable, this means we have already achieved the optimal range.

| Layer index | outlier fraction (%) |
|---|---|
| 1 | 0.3187 |
| 5 | 0.8579 |
| 10 | 0.3953 |
| 15 | 0.3975 |
| 20 | 0.3962 |
| 25 | 0.4399 |
| 30 | 0.3954 |

Table 8: Outlier fraction distribution in different layer index in BLOOM-7B under 3-sigma threshold

| steps | quantization range |
|---|---|
| 0 | 0.078 |
| 10 | 0.069 |
| 50 | 0.052 |
| 100 | 0.048 |
| 150 | 0.047 |
| 200 | 0.047 |

Table 9: The dynamic quantization range of different optimization steps. Here we take the quantization range of the Att.qkv module in layer 1 as an example.

# 6 Related Work

**Model Quantization** Traditional model quantization algorithms mainly focus on the cases where both parameters and activations of the model are quantized (Lin et al., 2015; Hubara et al., 2016; Tailor et al., 2021; Ni et al., 2020). However, directly quantizing the model will greatly decrease the accuracy of the models, and one important technique to improve the performance is Quantization Aware Training (QAT) (Jacob et al., 2018), where it simulates the quantization procedure in training to improve the accuracy of the quantized model further. For Transformer based models, the boundary of the compression level has been continuously advanced. For example, 8-bits quantized transformers as in FullyQT (Prato et al., 2019) and Q8BERT (Zafrir et al., 2019), 4-bits quantized BERT in Wu et al. (2023) and tenary case as in TernaryBERT (Zhang et al., 2020).

**Model Quantization for LLMs.** For quantizing LLMs, due to their prohibitive training expense, we can only use a few training data for calibration. There are two major directions: 1) weight-only quantization, where the weights are quantized into lower bits. In Frantar et al. (2023a); Yao et al. (2022), authors optimize the output error on the calibration set using OBS and gradient descent. 2)

Activation and weight quantization, where both activations and weights are quantized into lower bits. In this case, the major obstacle is the outliers in activations. LLM.int8() (Dettmers et al., 2022) addresses this problem by isolating those outliers in fp16/bf16. However, such implementation leads to large latency overhead and is even slower than fp16 inference. Recent studies (Wei et al., 2023; Xiao et al., 2023) found that the outliers only exist in certain channels, and use the LayerNorm weights (Wei et al., 2023) and calibrated scales (Xiao et al., 2023) to smooth those channels. Xiao et al. (2023) has already proved that we can achieve almost lossless W8A8 quantized LLMs using a few calibration data, without manipulating the original model weights.

## 7 Conclusion and Limitations

In this paper, we propose a data-free fast weight-only quantization algorithm, namely EasyQuant, for LLMs, that potentially improves the quantized model's performance without using any training data. Our analysis reveals the intrinsic origins of the performance loss when quantizing the model weights into lower bits. We show that by isolating the outliers from quantization, the accuracy of the quantized LLM increases accordingly with decreased reconstruction error. Our experiment proved that EasyQuant significantly outperforms RTN in a data-free setting, and also behaves better than data-dependent algorithms. EasyQuant can finish the quantization for a 176B-sized model within 10 minutes and the overhead of dequantization in EasyQuant is negligible.

However, we also point out some limitations of our work: The outlier recovery functionality in EasyQuant requires extra CUDA kernels for implementation. Moreover, weight-only quantization can only reduce the memory footprint without any computation cost reduction, hence the latency of our model cannot be minimized. In addition, this outlier isolation will make the weight/activation quantization more challenging because the weight includes numbers under different precision. We have also noticed that EasyQuantcannot outperform the data-dependent methods in all tasks, this motivates us to investigate more effective algorithms in future studies.

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

# A Appendix

| | | Perplexity-based Task | | | Zero-shot Task | | | |
|---|---|---|---|---|---|---|---|---|
| | | WikiText2 | PTB | C4 | PIQA | ARC-easy | ARC-Challenge | StoryCloze |
| BLOOM 560M | fp16 | 22.42 | 43.69 | 26.6 | 65.07% | 41.71% | 24.15% | 61.94% |
| | RTN | 25.90 | 51.10 | 29.89 | 63.11% | 39.40% | 23.89% | 60.15% |
| | GPTQ | 24.03 | 46.97 | **28** | **64.31**% | 40.24% | 23.46% | **61.17**% |
| | EasyQuant | **23.74** | **46.86** | 28.03 | 63.06% | **40.32**% | **24.15**% | 59.64% |
| BLOOM 1.1B | fp16 | 17.69 | 57.96 | 22.05 | 67.14% | 45.41% | 25.68% | 63.27% |
| | RTN | 22.00 | 66.85 | 24.44 | 65.29% | 42.51% | 23.34% | 60.66% |
| | GPTQ | 19.05 | 62.48 | 23.25 | 66.05% | **44.49**% | 25.51% | **62.32**% |
| | EasyQuant | **18.51** | **61.83** | **22.94** | **66.65**% | 43.73% | 25.51% | 62.06% |
| BLOOM 1.7B | fp16 | 15.39 | 30.00 | 19.49 | 69.97% | 48.11% | 26.79 % | 65.44% |
| | RTN | 16.97 | 33.58 | 21.26 | 67.74% | 44.70% | 26.45 % | 62.95% |
| | GPTQ | 16.48 | 31.84 | 20.55 | 68.77% | 44.49% | 25.94% | 64.48% |
| | EasyQuant | **16.01** | **31.50** | **20.15** | **68.99**% | **46.89**% | **26.19**% | **65.37**% |
| BLOOM 3B | fp16 | 13.48 | 25.34 | 17.49 | 70.51% | 53.24% | 30.55 % | 67.79% |
| | RTN | 14.76 | 27.68 | 18.76 | 69.86% | 51.35% | 29.52% | 67.09% |
| | GPTQ | 14.2 | 26.49 | 18.1 | 69.42% | **52.82**% | **28.92**% | 67.22% |
| | EasyQuant | **14.01** | **26.12** | **17.96** | **69.80**% | 50.72% | 28.58% | **67.35**% |
| BLOOM 7.1B | fp16 | 11.37 | 20.83 | 15.20 | 73.72% | 57.37% | 33.45 % | 71.99% |
| | RTN | 12.10 | 22.42 | 16.06 | 72.69% | 56.14% | 32.17 % | 70.72% |
| | GPTQ | 11.73 | 21.67 | 15.6 | 72.96% | **56.14**% | 32.25% | **71.36**% |
| | EasyQuant | **11.66** | **21.47** | **15.52** | **73.23**% | 55.72% | **32.51** % | 71.10% |
| BLOOM 176B | fp16 | 8.11 | 14.59 | 11.71 | 79.16% | 67.47% | 44.97 % | 76.89% |
| | RTN | 8.37 | 15.00 | 12.04 | 79.00% | 66.33% | 43.17 % | 76.00% |
| | GPTQ | 8.21 | 14.75 | **11.81** | 79.00% | 67.42% | 44.10% | 76.32% |
| | EasyQuant | 8.21 | 14.75 | 11.87 | **79.05**% | **67.8**% | **44.45**% | **77.28**% |

Table 10: Perplexity and zershot results for BLOOM model family