# OpenReview forum: "EasyQuant: An Efficient Data-free Quantization Algorithm for LLMs"
_EMNLP/2023/Conference — EMNLP 2023 Main_

### Official Review · Reviewer_uuQc · 2023-08-03

**Soundness:** 3

**Excitement:**

3: Ambivalent: It has merits (e.g., it reports state-of-the-art results, the idea is nice), but there are key weaknesses (e.g., it describes incremental work), and it can significantly benefit from another round of revision. However, I won't object to accepting it if my co-reviewers champion it.

**Paper Topic And Main Contributions:**

This paper proposes a data-free weight-only quantization method (EasyQuant) for large language models (LLMs), which aims to select the outliers in the weights by utilizing the mean and variance of weights, then isolate them from the quantization operation. Based on that, the quantization range is optimized using the gradient of the reconstruction error to quantize the remaining the normal values into lower bits. The extensive experiments validate the effectiveness of EasyQuant.

**Questions For The Authors:**

Answer the above questions.

**Reasons To Accept:**

(1) The first attempt to quantize the LLMs without accessing the input data.
(2) The proposed method reveals the impacts of the outliers in weights and the quantization range on the generalization of low-bits quantized LLMs.
(3) High quantization efficiency. For instance, EasyQuant can quantize public-available LLMs (e.g., OPT-176B, BLOOM-176B, and LLAMA-65B) into lower bits with a few minutes.


**Reasons To Reject:**

(1) The authors argue that the outliers in the weights have the non-negligible impacts on the quantization process. Hence, some crucial questions deserve to be studied: how does the outliers distribute in the weights? whether does they always exist? is the number (or percent) of the outliers still the same for different layers?
(2) EasyQuant tends to keep the outliers in full precision (fp32/fp16/bf16) while the normal values are quantized into low bits, which seems to be equivalent to the mix-precision quantization. Besides, it is unclear whether the performance advantages are from the valuable outliers that are kept in full precision.
(3) EasyQuant consider only the 4-bit case. When encountering lower bit width, will more outliers appear? Under such situation, can the quantization range be effectively optimized?
(4) The experiments are insufficient.
--- Instead of the fixed quantization range, it is expected to verify how the quantization scale s changes during the optimization, especially for different layers.
--- Lack the validation experiments about how the outlier threshold n affects the performance.


**Reproducibility:**

3: Could reproduce the results with some difficulty. The settings of parameters are underspecified or subjectively determined; the training/evaluation data are not widely available.

**Reviewer Confidence:**

4: Quite sure. I tried to check the important points carefully. It's unlikely, though conceivable, that I missed something that should affect my ratings.

---

> ### Author Rebuttal · Authors · 2023-08-29
>
> We appreciate your great efforts and constructive comments, and we have tried our best to address your about the outlier isolation design of our algorithm. Please let us know if you have any further questions.
> ### Q1. Distribution of outliers among different modules, layers, and models:
> In the Table below, we show that the fraction of outliers shares different pattern in different modules and layers; Moreover, we also find different models shares different fraction of outliers (BLOOM-176B admits 1% outliers while LLAMA-65B has about 0.2% outliers). We will add more discussion about this in our revised version.
> | module name | Att.qkv | Att.output | FFN.1 | FFN.2 |
> | -- | -- | -- | -- | -- |
> | outlier fraction (%) | 0.2993 | 0.5036 | 0.288 | 0.7560 |
> Table 1. Outlier fraction distribution in different modules in BLOOM-7B  under 3-sigma threshold
>
> | Layer index  | 1 | 5 | 10 | 15 | 20 | 25 | 30|
> | -- | -- | -- | -- | -- | -- | -- | -- |
> | outlier fraction (%) | 0.3187 | 0.8579 | 0.3953 | 0.3975 | 0.3962 | 0.4399 | 0.3954 |
> Table 2. Outlier fraction distribution in different layers   in BLOOM-7B under 3-sigma threshold
>
> ### Q2. Whether the performance gain mainly comes  from the outlier isolation:
> Actually, outlier isolation is a very important component of EasyQuant, but still not enough to fully recover the performance loss from quantization. Keeping even **10%**  of weights as fp16 outliers still admits about 8% ppl increase while EasyQuant admits only 1% ppl increase. Moreover, in Table 3 we show that outlier isolation somehow behaves like a **gating** mechanism for quantization range optimization: without outlier isolation, the model achieves a much worse performance under a small reconstruction error; however, when keeping those outliers in fp16, the quantized LLM attains a continuously decreased ppl under smaller reconstruction error. Therefore it is hard to judge which part is more important in EasyQuant.
>
> | reconstruction error | 4.8E4 | 3.5E4 | 2.7E4 | 2.3E4 |1.9E4 |
> |--|--|--|--|--|--|
> | int4 outlier--PPL | 12.65 | 14.73 |  19.71 | NA | NA |
> | fp16 outlier--PPL | 12.50 | 11.61 |  11.25 | 11.10 | 11.02 |
> Table 3. PPL results on Wikitext2 of BLOOM-7B with and without outlier isolation
>
> ### Q3. Under lower bits case, will more outliers be needed and will EasyQuant still work:
> For 3-bit quantization, EasyQuant still significantly (more than 10%) outperforms both RTN and GPTQ (which is data-dependent) on LLAMA-7B using 3-sigma threshold (about 0.2% outliers), which is the same setting in 4-bit quantization:
> | benchmark (PPL) | RTN-3bits | GPTQ-3bits | EasyQuant-3bits|
> |--|--|--|--|
> | Wikitext2 | 25.54 | 8.07 | 7.16 |
> Table 4. Comparison of different methods under 3-bit quantization
>
> Since the outliers are the salient values in weights, we believe that for lower-bit quantization, we can still use the same fraction of outliers as it is in 4-bits. We will add more experiments for 3-bit quantization of EasyQuant on more benchmarks in our revised version.
>
> ### Q4. More experiments about the dynamic of the quantization range:
> The dynamic of the quantization range is shown in Table 4. Roughly speaking, this range decreases fast in the early stage of training, which means a smaller quantization range will make most of the parameters to be quantized more precisely. After certain steps of training, the quantization range becomes stable, this means we have already achieved the optimal range.
>
> | steps  | 0 |  10 | 30 |50 | 100 | 150 | 200|
> | -- | -- | -- | -- | -- | -- | -- | -- |
> | quantization range  | 0.078 | 0.069 | 0.056 | 0.052 |  0.048 | 0.047 | 0.047 |
>
> Table 4. The dynamic quantization range of different optimization steps. Here we take the quantization range of the Att.qkv module in layer 1 as an example.
>
>
> ### Q5. How does the outlier threshold n affect the performance:
> | threshold | 5 | 4 | 3 | 2 | 1 |
> | -- | -- | -- | -- | -- | -- |
> | ppl on Wikitext2 | 6.20 | 6.10 | 5.98 | 5. 89 | 5.90 |
> | ppl on PTB | 11.02 | 10.86 | 10.68 | 10.40 | 10.50 |
> Table 5. LLAMA-7B ppl results under different outlier thresholds.
>
> As shown in Table 5, increasing the outlier fraction is beneficial to the quantized model, choosing threshold=3 is to balance the outlier overhead and model performance. Increasing the outlier fraction (under acceptable overhead) is an efficient method to improve the performance of quantized LLMs.

---

### Official Review · Reviewer_1vKT · 2023-08-10

**Soundness:** 4

**Excitement:**

4: Strong: This paper deepens the understanding of some phenomenon or lowers the barriers to an existing research direction.

**Paper Topic And Main Contributions:**

This paper proposes an efficient data-free quantization algorithm called EasyQuant for model compression. The key contributions are:

· EasyQuant is a training-free and data-free weight-only quantization method. It has the advantage of guaranteeing the generalization ability of the quantized model, which is important for universal models like LLMs.
· The paper finds through experiments that the two main reasons for performance degradation are suboptimal quantization ranges and improper handling of outliers in weights. EasyQuant addresses this by excluding outliers and optimizing the quantization ranges.
· Compared with data-dependent methods, EasyQuant achieves better or comparable accuracy while ensuring generalization and runs over 10x faster.
· EasyQuant is implemented in a highly parallel way on GPUs, enabling quantization of models with over 100B parameters within minutes. This makes it practical for large models.

The paper provides useful insights on the importance of optimizing quantization ranges and handling outliers, informing future research.
In summary, EasyQuant presents an effective data-free approach for low-bit quantization of LLMs, achieving good accuracy and efficiency. It also sheds light on important principles in model quantization. The results and insights are valuable and novel contributions to the field of model compression.

**Reasons To Accept:**

The paper makes an important contribution by revealing that outliers in weights and optimization of quantization ranges are essential factors for reducing quantization error. Specifically, the key conclusions are: Through experiments, the paper finds that directly shrinking the quantization range will clip most outliers, severely degrading model performance even as reconstruction error decreases. This counterintuitive finding underscores the importance of outliers. Keeping outliers (e.g. less than 1% of weights) at full precision alone is not sufficient to recover accuracy - quantization ranges must also be optimized. By leaving outliers unchanged and optimizing quantization ranges via gradient descent, EasyQuant is able to significantly improve accuracy versus baseline quantization methods that do not account for these factors.

**Reasons To Reject:**

No

**Reproducibility:**

4: Could mostly reproduce the results, but there may be some variation because of sample variance or minor variations in their interpretation of the protocol or method.

**Reviewer Confidence:**

3: Pretty sure, but there's a chance I missed something. Although I have a good feel for this area in general, I did not carefully check the paper's details, e.g., the math, experimental design, or novelty.

---

> ### Author Rebuttal · Authors · 2023-08-29
>
> We sincerely appreciate your affirmation of our work. Yes, our method consists of two parts: outlier isolation and range optimization. EasyQuant is indeed the first data-independent LLM quantization algorithm that outperforms the previous data-dependent SOTA method. Moreover, EasyQuant will not make the model overfit to certain tasks since it uses no calibration data, which means the generalization ability of quantized LLMs can be safely guaranteed. Please let us know if you have any further questions.

---

### Official Review · Reviewer_Mg7b · 2023-08-11

**Soundness:** 3

**Excitement:**

2: Mediocre: This paper makes marginal contributions (vs non-contemporaneous work), so I would rather not see it in the conference.

**Paper Topic And Main Contributions:**

This paper presents a data-free quantization method specifically designed for addressing outliers in Large Language Models (LLMs). By separately quantizing the non-outlier portion of the weights, the method aims to mitigate the performance drop caused by outliers. The authors test their approach on two model families, LLaMAs and BLOOMs, using INT4 bit for both Language Model tasks and zero-shot QA tasks. The results demonstrate a slight gain in performance for some tasks and models compared to previous methods. Additionally, the authors evaluate the overhead of their proposed data-free quantization build.

**Questions For The Authors:**

**Question1**: Is there more evidence to support that the performance drop in lower-bit quantization is due to weight outliers, such as experiments on various tasks and analysis of weight outlier distributions?

**Question2**: Are there any comparisons between the proposed method and the two state-of-the-art baselines, LLM.int8() and SmoothQuant?

**Reasons To Accept:**

1. The motivation behind the proposed approach is reasonable.
2. The authors have chosen to test their method on more sensitive language model tasks, and some results outperform the baselines.

**Reasons To Reject:**

1. In the quantization community, the discussion on outliers mainly focuses on activation outliers, such as SmoothQuant and LLM.int8(). It would be helpful to provide more evidence to support the claim that the performance drop in lower-bit quantization is due to weight outliers, for example, by conducting experiments on various tasks and analyzing the distribution of weight outliers.
2. The motivation is relatively straightforward, the paper is closely related to works such as LLM.int8(), which may result in lower novelty.
3. The writing of this paper is somewhat difficult to understand. For instance, the introduction does not mention any technical details, and the core contribution is not discussed until Section 4.2.
4. The experimental section has relatively weak baselines, as it does not compare the proposed method with LLM.int8() or SmoothQuant. Additionally, the performance improvements brought by the proposed method are minimal.

**Reproducibility:**

3: Could reproduce the results with some difficulty. The settings of parameters are underspecified or subjectively determined; the training/evaluation data are not widely available.

**Reviewer Confidence:**

4: Quite sure. I tried to check the important points carefully. It's unlikely, though conceivable, that I missed something that should affect my ratings.

**Typos Grammar Style And Presentation Improvements:**

The formatting of the tables is somewhat disorganized, particularly with Table 3 being excessively large. It should have a dedicated section for discussing the limitations instead of in the conclusion.

---

> ### Author Rebuttal · Authors · 2023-08-29
>
> We appreciate your great efforts and constructive comments, and we have tried our best to address your concerns about the novelty and design of our algorithm. Please let us know if you have any further questions.
>
> ### Q1. Lack of novelty compared to LLM.int8() and SmoothQuant:
> Here we politely point out that LLM.int8() and SmoothQuant are orthogonal to our work because their method cannot work on low-bits weight-only quantization (the ppl becomes larger than 1E3 on 4-bit quantized LLAMA-7B). We emphasize that the novelty of our work comes from three parts:
> 1.  Our work points out that the importance of outliers in weights (not activations) is underestimated, and those outliers impose a **gating** mechanism (see our answer to Q3 below) instead of a direct influence. This is a critical observation in low-bits LLM quantization that has not been found before.
> 2. Outlier isolation is only one part of our work. Beyond this, we propose a fast method for updating the quantization range, which essentially helps EasyQuant to achieve superior performance compared to the previous SOTA method (Please refer to Table 1 in our paper).
> 3. This is the first efficient data-independent LLM quantization algorithm that achieves a better performance than the data-dependent algorithms. The data-independent nature of EasyQuant ensures that the generalization ability of quantized LLMs is safely guaranteed. We argue that this is a **very important** requirement for LLM quantization and should not be underestimated.
>
> ### Q2. The performance  improvements brought by the proposed method are minimal:
> As shown in Table 2,3 in our paper, EasyQuant outperforms the data-independent baseline (RTN) on almost all benchmarks. For BLOOM-1.7B, it achieves 5%\~10% relative ppl drop and 1%\~3% absolute accuracy increase. For 3-bit quantization, EasyQuant significantly (more than 10%) outperforms both RTN and GPTQ (which is data-dependent) on LLAMA-7B:
> | Algorithm | RTN-3bits | GPTQ-3bits | EasyQuant-3bits|
> |--|--|--|--|
> | Wikitext2 | 25.54 | 8.07 | 7.16 |
> Table 1. Comparison of different methods under 3-bit quantization
>
> Therefore we politely point out that the performance gain is not trivial.
>
> ### Q3. Evidence for the importance of the outliers in weights:
> The outlier isolation is a key component in EasyQuant, but it can only impose an indirect influence on the model accuracy (as shown in Figure 1 of our paper). The interesting phenomenon we find is that the outliers behave like a **gating** mechanism:  without outlier isolation, the model achieves a much worse performance under a small reconstruction error; however, when keeping those outliers in fp16, the quantized LLM attains a continuously decreased ppl under smaller reconstruction error:
> | reconstruction error | 4.8E4 | 3.5E4 | 2.7E4 | 2.3E4 |1.9E4 |
> |--|--|--|--|--|--|
> | int4 outlier | 12.65 | 14.73 |  19.71 | NA | NA |
> | fp16 outlier | 12.50 | 11.61 |  11.25 | 11.10 | 11.02 |
> Table 2. ppl results on Wikitext2 of BLOOM-7B with and without outlier isolation
>
> Moreover, we have also conducted a complementary experiment testing the direct influence of the weight outlier: We prune 1% of the values ( according to its magnitude) in weights into 0 and see the ppl results:
> | pruned weights | smallest (top-0%\~1%) | median (top-49%\~50%) | largest (top-99%\~100%) |
> |--|--|--|--|
> | PPL | 11.66 | 19.16 |  19.17 |
> Table 3. ppl results after pruning 1% weight with different magnitude
>
> We can see that the largest value (outliers) imposes the same influence on the model performance as the normal values (median), which means those outliers share the same direct influence on the model accuracy with normal values. Therefore outlier isolation imposes a key influence on the model accuracy indirectly.
>
> ### Q4. More experiments proving whether the performance gain mainly comes from the outlier isolation:
> Actually, outlier isolation is a very important component of EasyQuant, but still not enough to fully recover the performance loss from quantization. Keeping even **10%** of weights as fp16 outliers still admits about 8% ppl increase while EasyQuant admits only 1% ppl increase. Below we present the result of 4-bit quantized BLLOM-7B when we just keep 1% outliers in fp16 without quantization range optimization on various benchmarks.
>
> | Benchmark | WikiText2  (PPL-based $\downarrow$) | PTB  (PPL-based $\downarrow$) | C4   (PPL-based $\downarrow$)  | PIQA  (ACC-based $\uparrow$) |
> | -- | -- | -- | -- | -- |
> | EasyQuant | 11.01 | 21.42  | 15.46 | 73.61% |
> | 1% fp16 outlier | 12.52 |  23.32 | 16.44  | 72.74% |
>
> Table 4. Using outlier isolation solely is not enough to fully recover the performance loss. EasyQuant consistently outperforms outlier isolation in all benchmarks.
>
> ### Q5. Distribution of the outliers in weights:
> We find that the fraction of outliers shares different patterns in different modules and layers (as shown in Table 5 and 6). We will add more discussion about this in our revised version.
> | module name | Att.qkv | Att.output | FFN.1 | FFN.2 |
> | -- | -- | -- | -- | -- |
> | outlier fraction (%) | 0.2993 | 0.5036 | 0.288 | 0.7560 |
> Table 5. Outlier fraction distribution in different modules in BLOOM-7B under 3-sigma threshold
>
> | Layer index  | 1 | 5 | 10 | 15 | 20 | 25 | 30|
> | -- | -- | -- | -- | -- | -- | -- | -- |
> | outlier fraction (%) | 0.3187 | 0.8579 | 0.3953 | 0.3975 | 0.3962 | 0.4399 | 0.3954 |
> Table 6. Outlier fraction distribution in different layers in BLOOM-7B under 3-sigma threshold
>
>
>
>
> In summary, we politely point out LLM.int8() and SmoothQuant are designed for activation quantization and are not fair baselines for comparison, as they work poorly in low-bits weight quantization. Our work reveals the importance of the weight outliers and proposes an efficient method (outlier isolation+range optimization) to overcome the performance drop in LLM quantization. The intrinsic data-independent nature of our method shall be appreciated because it ensures the generalization property of LLMs.

---

### Meta-Review · Area_Chair_c2CR · 2023-09-26

**Recommendation:** 4

**Metareview:**

The paper proposes a quantization technique independent of training data. Reviewers see merit in the proposal, especially because it is the first data-independent quantization technique. Reviewers have some outstanding questions (primarily about outliers).

---

### Decision · Program_Chairs · 2023-10-07

**Decision:**

Accept-Main

**Comment:**

The paper proposes a quantization technique independent of training data. Reviewers see merit in the proposal, especially because it is the first data-independent quantization technique. Reviewers have some outstanding questions (primarily about outliers).